# A Systems Biology Approach to Investigating the Interaction between Serotonin Synthesis by Tryptophan Hydroxylase and the Metabolic Homeostasis

**DOI:** 10.3390/ijms22052452

**Published:** 2021-02-28

**Authors:** Suhyeon Park, Yumin Kim, Jibeom Lee, Jeong Yun Lee, Hail Kim, Sunjae Lee, Chang-Myung Oh

**Affiliations:** 1Department of Biomedical Science and Engineering, Gwangju Institute of Science and Technology, Gwangju 61005, Korea; suhyeonpark78@gmail.com (S.P.); dbals123@gm.gist.ac.kr (Y.K.); cocacola@gist.ac.kr (J.L); joyun321@gist.ac.kr (J.Y.L.); 2Graduate School of Medical Science and Engineering, Korea Advanced Institute of Science and Technology, Daejeon 34141, Korea; hailkim@kaist.edu; 3Department of School of Life Sciences, Gwangju Institute of Science and Technology, Gwangju 61005, Korea

**Keywords:** serotonin, metabolic homeostasis, systems biology

## Abstract

Obesity has become a global public health and economic problem. Obesity is a major risk factor for a number of complications, such as type 2 diabetes, cardiovascular disease, fatty liver disease, and cancer. Serotonin (5-hydroxytryptamine [5-HT]) is a biogenic monoamine that plays various roles in metabolic homeostasis. It is well known that central 5-HT regulates appetite and mood. Several 5-HT receptor agonists and selective serotonin receptor uptake inhibitors (SSRIs) have shown beneficial effects on appetite and mood control in clinics. Although several genetic polymorphisms related to 5-HT synthesis and its receptors are strongly associated with obesity, there is little evidence of the role of peripheral 5-HT in human metabolism. In this study, we performed a systemic analysis of transcriptome data from the Genotype-Tissue Expression (GTEX) database. We investigated the expression of 5-HT and tryptophan hydroxylase (TPH), the rate-limiting enzyme of 5-HT biosynthesis, in the human brain and peripheral tissues. We also performed differential gene expression analysis and predicted changes in metabolites by comparing gene expressions of tissues with high TPH expression to the gene expressions of tissues with low TPH expression. Our analyses provide strong evidence that serotonin plays an important role in the regulation of metabolic homeostasis in humans.

## 1. Introduction

Obesity is defined as excessive accumulation of body fat, which presents several risks [1]. It is the major risk factor for a number of complications such as type 2 diabetes, cardiovascular disease, fatty liver disease, and cancer [2]. The prevalence of obesity has increased dramatically and has become a global public health problem [2]. For this reason, the American Medical Association (AMA) recognized obesity as a complex, chronic disease [3]. For the prevention and treatment of this disease, scientists have tried to understand the mechanisms of regulating energy homeostasis and find a way to maintain a balance between energy intake and energy expenditure [4,5].

Serotonin (5-hydroxytryptamine, 5-HT) is a biogenic monoamine that is highly conserved between nematodes and vertebrates [6]. 5-HT is synthesized from tryptophan catalyzed by the rate-limiting enzyme, tryptophan hydroxylase (TPH) [7]. TPH exists in two isoforms, TPH1 and TPH2. TPH1 is mainly expressed in peripheral tissues and pineal gland [8]. TPH2 is abundant in the serotonergic neurons in the brain and myenteric plexus [9]. Recent studies have revealed various roles of serotonin in the regulation of energy homeostasis [7,10,11]. In the brain, serotonin controls anxiety and appetite-related behaviors as a neurotransmitter [10]. Several serotonin receptor agonists and selective serotonin receptor uptake inhibitors (SSRIs) targeting central serotonin have been widely used in the clinical field and have revealed strong associations with body weight changes [10]. For example, serotonin receptor (HTR) 2C is a G protein-coupled receptor that plays a role in appetite, eating behavior, and energy metabolism [7,12]. HTR2C polymorphisms have also shown strong associations with obesity and metabolic disorders [13,14], and the HTR2C agonist has shown a significant weight-loss effect in clinical studies [15].

Peripheral serotonin also has shown integral roles in various physiological and pathological regulation [7,10,11]. More than 90% of peripheral serotonin is synthesized and secreted from enterochromaffin cells in the gut and stored in platelets [7]. Platelets uptake 5-HT from the plasma and release 5-HT in response to specific conditions such as tissue injury and acute inflammation [16,17]. Furthermore, gut-derived serotonin (GDS) promotes gluconeogenesis and lipolysis in hepatocytes under fasting conditions [18]. GDS also plays an important role in lipid accumulation in the liver. Inhibition of GDS synthesis or HTR2A signaling prevents high-fat diet (HFD) induced hepatic steatosis in mice models [19]. Serotonin derived from other peripheral tissues has shown various effects on metabolism [20,21,22,23]. Pancreatic beta cell-derived serotonin regulates beta cell proliferation and insulin secretion during pregnancy [20,21]. Adipocyte-derived serotonin (ADS) regulates lipogenesis in white adipose tissue (WAT) and thermogenesis in brown adipose tissue (BAT) [22,23]. Recent studies have reported that the gut microbiome controls GDS synthesis and serotonin level changes due to gut microbiota dysbiosis results in obesity and its related metabolic dysfunctions [24,25].

Genetic studies have suggested statistical evidence for the role of serotonin in metabolic disorders in humans [26,27,28,29]. Recent monozygotic study reported that serotonin transporter gene (SCL6A4) promoter hypermethylation has strong association with body weight and body mass index (BMI) [26]. Genetic polymorphisms and DNA hypermethylation of the HTR2A gene have been associated with obesity and metabolic syndrome [27]. Single-nucleotide polymorphisms (SNPs) in HTR2A and HTR2C have significant associations with obesity and type 2 diabetes [28]. SNPs in TPH1 and HTR2B displayed significant associations with weight gain during pregnancy [29].

Animal studies and human genomic studies have demonstrated that serotonin regulates glucose, lipid metabolism, and energy expenditure [7,11]. Although some human studies reported that plasma 5-HT had associations with obesity [30,31], more evidence is needed to support the role and potential mechanisms of 5-HT in human peripheral tissues. In this study, we aimed to analyze the gene expressions of the human brain and peripheral tissues according to TPH expression. To accomplish this, we used the Genotype-Tissue Expression (GTEX) database [32].

## 2. Results

### 2.1. TPH Expression in Human Tissue

TPH is the rate-limiting enzyme in the biosynthesis of serotonin. In the brain, serotonin levels are directly related to TPH activity [33]. In peripheral tissues, serotonin levels are related to circulating serotonin levels from the gut (circulating serotonin) and TPH activity of the peripheral tissues (local serotonin) [34]. To assess the roles of serotonin in energy metabolism, we analyzed TPH mRNA expression in the human tissues from the GTEx dataset. First, we checked TPH1 and TPH2 expression in the brain and peripheral tissues (Figure 1 and Appendix A). TPH2 was mostly expressed in central nervous system (Figure 1A). Gastrointestinal tracks expressed TPH1, and several peripheral tissues, such as adipose tissues, expressed TPH1 (Figure 1B). These expressions support previous reports about the existence and autocrine/paracrine role of ADS as well as GDS [19,22].

Previous postmortem analysis of brain tissues reported that TPH2 mRNA expression is abundant in the dorsal raphe nucleus, median raphe nucleus, and raphe nuclei-containing regions such as pons and medullar, not the pituitary gland [35,36]. Although TPH1 is mainly expressed in peripheral tissues, some papers have already reported the existence of TPH1 in the brain, especially in the pituitary gland [37,38,39]. Our results showed that the pituitary gland expressed TPH1 as well as TPH2, and the expression levels were higher than those in other tissues (Figure 1). This suggests that serotonin might play a more important role in the pituitary gland and hypothalamus–pituitary axis regulation than we already thought.

### 2.2. Transcriptome Analysis in Brain According to TPH2 Expression

To investigate the role of serotonin in the brain, we selected 10 brain tissues with highly expressed TPH2 mRNA and 10 brain tissues with lowly expressed TPH2 mRNA in the brain transcriptome of GTEx dataset. Differentially expressed gene (DEG) analysis between these two groups revealed that high TPH2 groups showed downregulation of lipid metabolism related genes (Figure 2A,B). For example, adipogenesis marker genes (FAB4 and ADIPOQ) and driver gene (PPARG) are decreased in high TPH2 group [40]. Fatty acid oxidation (FAO) related genes (CIDEA, UCP2, ANGPTL4) are also decreased in high TPH2 group [41]. Gene Set Enrichment assay (GSEA) revealed that central serotonin has a negative association with fatty acid metabolism, adipogenesis, and glycolysis (Figure 2C,D). Lipid in the brain is a key component of neuronal structure and brain development [42]. Thus, our results imply that serotonin has a significant role in lipid processing in the brain, which regulates systemic metabolism [42]. Gene Ontology (GO) analysis supported this implication. GO analysis revealed that several biological functions associated with sensory perception and neuronal development changed according to TPH2 expression in brain (Figure 2E).

### 2.3. Transcriptome Analysis in Intestine and Adipose Tissue According to TPH1 Expression

The major source of peripheral serotonin is enterochromaffin cells in the gut [7]. This GDS directly regulates intestinal motility and inflammation [43]. As a circulating hormone, GDS inhibits bone formation [44] and regulates lipid metabolism in the adipose tissue and liver [18,19]. To evaluate the role of serotonin in the gut, we selected 10 samples with highly expressed Tph1 mRNA and 10 samples with lowly expressed Tph1 mRNA from the small intestine and colon transcriptome data from the GTEx database. Figure 3 and Appendix A show highly differentially expressed genes from the small intestine and colon dataset. Intriguingly, most highly expressed genes in the high TPH1 group have important roles in pancreatic endocrine cell development (PDX1, PAX4, NEUROD1, NEUROG3) and hormonal secretion (CHGA, SST, DPP4) (Figure 3A,B). GSEA also showed that TPH1 expression was positively correlated with pancreatic beta cell related genes (Figure 3C,D). Previously, our group reported that pancreatic beta cell-derived serotonin regulates beta cell proliferation and insulin secretion [20,45]. However, the role of GDS in pancreatic beta cell has not yet been discovered. Further studies are needed to investigate the role of GDS in pancreatic beta cell. Conversely, these patterns suggest the importance of GDS in gut functions as an endocrine organ. Actually, gut is the largest endocrine organ in the human body [46], and many genes related to pancreatic beta cell also play an important role in gut endocrine functions. For example, Pdx1 deletion in the gut resulted in a significant reduction of mRNA abundance of gastric inhibitory peptide and somatostatin and decreased intestinal alkaline phosphate activity in the mouse gut [47]. Ngn3 deleted mice showed impaired endocrine progenitor cells, gastrin-secreting cells (G cells), and somatostatin-secreting cells (D cells) in the gut [48].

GSEA analysis also showed a positive correlation with fatty acid metabolism and the K-RAS signaling pathway (Figure 3C,D). KRAS signaling plays various roles in cell proliferation, apoptosis, and angiogenesis [49]. This gene is a well-known oncogene [50]. Mutations in this gene and activated KRAS signaling are one of the most common causes of colon cancer development [50,51]. This suggests that high serotonin levels in the gut may act as a cancer driver. Some studies have reported the role of serotonin in colon cancer [52,53,54]. Tutton et al. reported that serotonin supplementation promoted the proliferation of colon cancer cells [53]. GO analysis showed several pathways related to nutrient metabolism, organ development, and inflammation (Figure 3E).

### 2.4. Transcriptome Analysis in White Adipose Tissue According to TPH1 Expression

ADS regulates adipocyte differentiation and metabolism via autocrine/paracrine signaling [55]. Previously, we reported that serotonin regulates de novo lipogenesis in white adipose tissue and thermogenesis in brown adipose tissue in mice [22,23]. To evaluate the role of ADS in human adipose tissue, we selected 10 samples with highly expressed Tph1 mRNA and 10 samples with lowly expressed Tph1 mRNA from white adipose tissues (omentum and subcutaneous tissue beneath the leg’s skin) transcriptome data from the GTEx database. Figure 4 shows the results of DEG, GSEA, and GO analysis. The high TPH1 expression group showed significant metabolism-associated gene expression changes compared to the low TPH1 expression group (Figure 4A,B). RNASE13 is a ribonuclease, and this gene, which has the highest fold changes in GSEA, has shown significant associations with diabetes [56]. The transcription factor SIM1 plays a role in appetite control, and genetic variations in the SIM1 genes have shown significant associations with pediatric obesity [57]. GSEA analysis revealed that TPH1 expression in adipose tissue was positively correlated with pancreatic beta cell specific genes, fatty acid metabolism, and KRAS signaling (Figure 4C,D). Previously, we reported that pancreatic beta cell-derived serotonin regulates beta cell proliferation and insulin secretion [20,45,58]. However, there are no reports about the role of ADS in pancreatic beta cell. High TPH1 adipose tissues showed decreased KIRREL2 mRNA expression. KIRREL2, a novel immunoglobulin superfamily gene, is primarily expressed in pancreatic beta cells and regulates insulin secretion [59,60]. Further studies are needed to explore the role of GDS in pancreatic beta cell based on these changes in gene set analysis. GO analysis (Figure 4E) revealed the immune systems and organ development. Interestingly, developmental categories in other peripheral tissues such as lung, kidney, and pancreas have significant associations with TPH1 expression in adipose tissue. This suggests an important endocrine role of adipose tissue in the human body.

### 2.5. The Role of Serotonin in Mitochondria

Animal studies have revealed that serotonin is an important regulator of mitochondrial function [61,62,63]. Serotonin increases mitochondrial biogenesis through HTR2A in cortical neurons and regulates mitochondrial transport in hippocampal neurons through HTR1A [61]. HTR3 and HTR4 localize to the mitochondrial membrane and regulate mitochondrial functions by Ca^2+^ signaling and mitochondrial permeability transition pore (mPTP) opening [63]. Serotonin also acts as an antioxidant in brain and peripheral tissues, such as pancreatic beta cell and kidney [58,64,65]. Caenorhabditis elegans studies reported that serotonin is required for neuroendocrine communication against mitochondrial proteotoxic stress [66,67].

To investigate the role of serotonin in human mitochondria, we analyzed the expressions of genes related to mitochondrial biogenesis and quality control in human tissue transcriptome from the GTEx dataset. Figure 5 shows a heatmap plot of the DEG analysis. In the brain, TPH2 expression showed negative associations with most genes related to mitochondrial biogenesis, ATP biosynthesis, and mitochondrial quality control.

### 2.6. Metabolite Changes According to TPH Expression

Serotonin is the main tryptophan metabolite by TPH and is metabolized to 5-hdryoxyindole acetaldehyde by monoamine oxidase [8]. Numerous studies have shown that gut microbes can metabolize tryptophan in the gut and affect host metabolism by changing host tryptophan metabolites [25,68]. Yano et al. reported that indigenous spore-forming bacteria increase serotonin synthesis from colon enterochromaffin cells (ECs) [68]. Colons of germ-free mice showed decreased *Tph1* expression compared to *Tph1* expression in specific pathogen-free mice colon [68]. Serotonin also directly regulates gut microbiota composition [69].

In order to estimate metabolite changes, based on enzyme gene expression, in the gut according to serotonin levels, we performed reporter metabolite analysis, together with metabolic pathway analysis, using R-package piano [70]. The predicted metabolic changes are shown in Appendix A. Väremo L. et al. developed this gene set analysis method for the interpretation of metabolic and biological functions and pathways from microarray and RNA sequencing data [70]. Figure 6 shows the predicted metabolic changes according to TPH expression in human tissues. TPH1 expression was positively associated with several pathways related to hormone metabolism, xenobiotic metabolism, and nutritional signaling (Figure 6).

In addition, we also analyzed metabolic profiles in adipose tissue and brain (Figure 6). In the adipose tissues, TPH1 expression shows positive associations with hormonal metabolism and negative associations with mitochondrial FAO pathways. Mitochondrial FAO is the main metabolic pathway against lipid overload in adipose tissue [71]. Enhancing mitochondrial FAO is an emerging therapeutic strategy for obesity treatment [72,73]. ADS increases lipid accumulation in white adipose tissue. Negative correlation between TPH1 and mitochondrial FAO in human adipose tissues bears out that inhibiting serotonin synthesis in adipose tissue can be a new therapeutic candidate for obesity therapy [22].

GSA results in the brain displayed that TPH2 has significant associations with mitochondrial FAO pathways, amino acid metabolism, and nucleotide metabolism (Figure 6). These metabolic profiles suggest that serotonin plays an important role in brain development and energy metabolism [74,75]. Clinical studies have already reported the role of serotonin in brain development [76]. Serotonin is involved in neural crest stem cell regulation and is a critical factor in cell survival, growth, differentiation, and synaptogenesis [76]. Alterations in serotonin signaling at an early age resulted in behavior and metal health problems throughout the life span [77].

## 3. Discussion

In this study, we aimed to elucidate the role of serotonin in metabolic changes in human tissues. Here, we used transcriptomic data from the GTEx project [32]. Central serotonin in the human brain regulates fatty acid metabolism and mitochondrial biogenesis and quality control. In addition, central serotonin has a significant association with nucleotide metabolism, which is an important component of neuronal development.

Several studies have reported the protective role of 5-HT in mitochondrial dysfunction. 5-HT increased mitochondrial biogenesis in rodent cortical neurons [61] and mediates mitochondrial stress response in the neurons of C. elegans model [66]. Intriguingly, our data showed that high TPH2 group show lower expression of genes related to mitochondrial functions compared to low TPH2 group (Figure 5). This result implies TPH2 increase in brain could be the result of protective response against mitochondrial dysfunction. Further studies are needed to clarify this association between TPH2 and mitochondrial function.

TPH1 expressions in small intestine and colon have significant associations with hormone and nutrition regulation. This implies that GDS might be the major regulator of endocrine function and metabolic homeostasis in the gut. TPH1 expression in white adipose tissue shows significant associations with mitochondrial FAO and development pathways of other tissues such as the lung and kidney. These results indicate that ADS is a critical factor for endocrine function in white adipose tissue.

Our study has several limitations. First, we did not obtain serotonin levels in human tissues. Thus, we assumed that mRNA expression of TPH might reflect the level of serotonin. If we can directly measure serotonin levels in human tissue, it might be the best way to understand the role of serotonin. Unfortunately, serotonin measurement is very difficult. Serotonin levels in blood and tissue are very low and are rapidly metabolized by monoamine oxidase in living organs [78,79,80]. Second, we used predicted metabolic profiles by using the GSA method for the evaluation of metabolic pathway changes related to serotonin. Further studies are needed to confirm this result by direct measurement of metabolites in human tissues. Third, we used both omental adipose tissues and subcutaneous adipose tissues when we analyzed white adipose tissue transcriptome. This heterogeneity may act as a confounding factor [81].

In conclusion, our study provides strong evidence that serotonin plays significant roles in critical pathways such as mitochondrial homeostasis, energy metabolism, and organ development.

## 4. Material and Methods

### 4.1. Data Collection

Human brain and peripheral tissue transcriptome data reported in the GTEx Analysis release V8 (dbGap Study Accession: phs000424.v8.p2) were downloaded from the GTEx portal (www.gtexportal.org (accessed on 10 December 2020)) [82]. This dataset includes raw count values that were normalized using the DESeq package in R software.

We also obtained transcripts per million (TPM) values of the GTEx dataset from a public repository, the Human Protein Atlas (http://www.proteinatals.org/about/download (accessed on 10 December 2020)). We then estimated the TPH1 and TPH2 expression levels of each human tissue transcriptome.

### 4.2. Data Analysis

We selected the 10 highest and the 10 lowest TPH expression groups from brain, small intestine, colon, and adipose tissue. We then compared gene expression and pathways of interest between the high and low TPH groups. DEGs were identified using the DESeq package in R software. Appendix A lists the DEGs of our analysis. Volcano plots and heatmaps were obtained using the R ggplot and gplots package.

The gene set–gene annotation database v7.2 was obtained from the GSEA Molecular Signature Database (https://www.gsea-msigdb.org/gsea/msigdb (accessed on 10 December 2020)). Then, gene set enrichment analysis (GSEA) was performed for gene set association analysis by using the GSEA software [83]. Gene ontology (GO) mining and pathway analysis were performed by the DAIVD gene functional classification tool [84]. Reporter metabolite analysis was performed using the Piano package [70]. The criteria for FDR-adjusted *p* values <0.05 were considered significant gene sets.

## Figures and Tables

**Figure 1 ijms-22-02452-f001:**
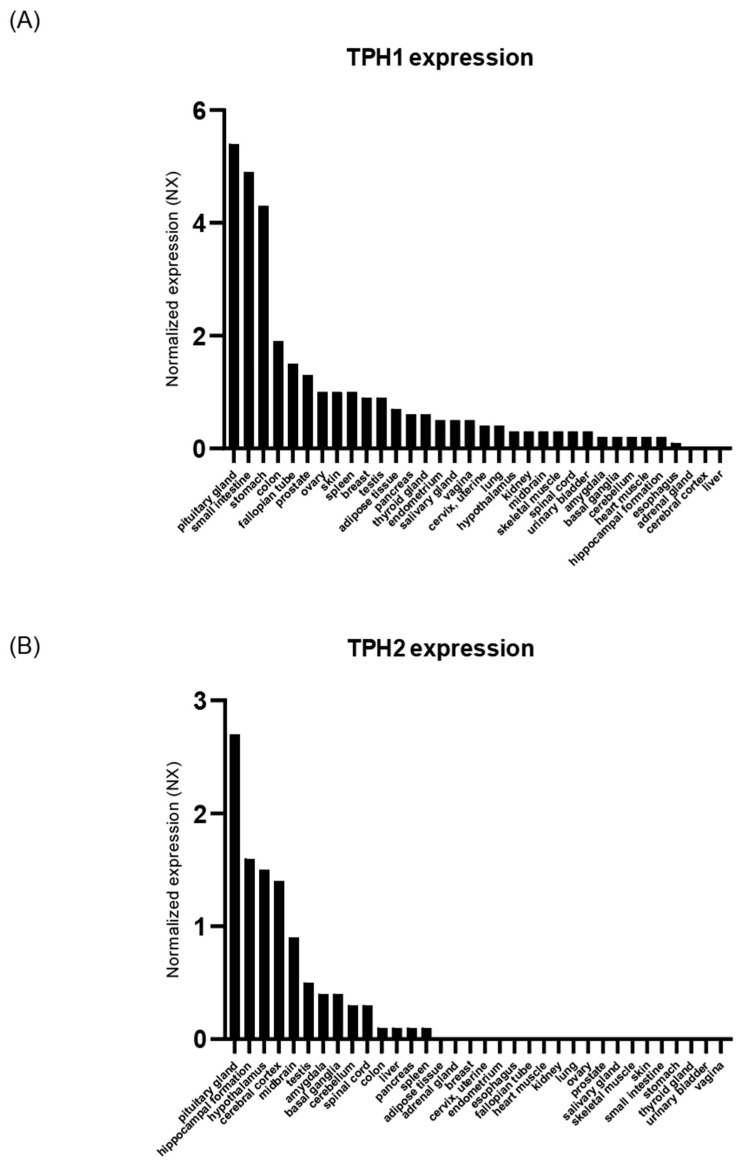
TPH expression in human tissues of the GTEx data. (**A**) TPH1 expression in human tissues. (**B**) TPH2 expression in human tissues. TPH: tryptophan hydroxylase.

**Figure 2 ijms-22-02452-f002:**
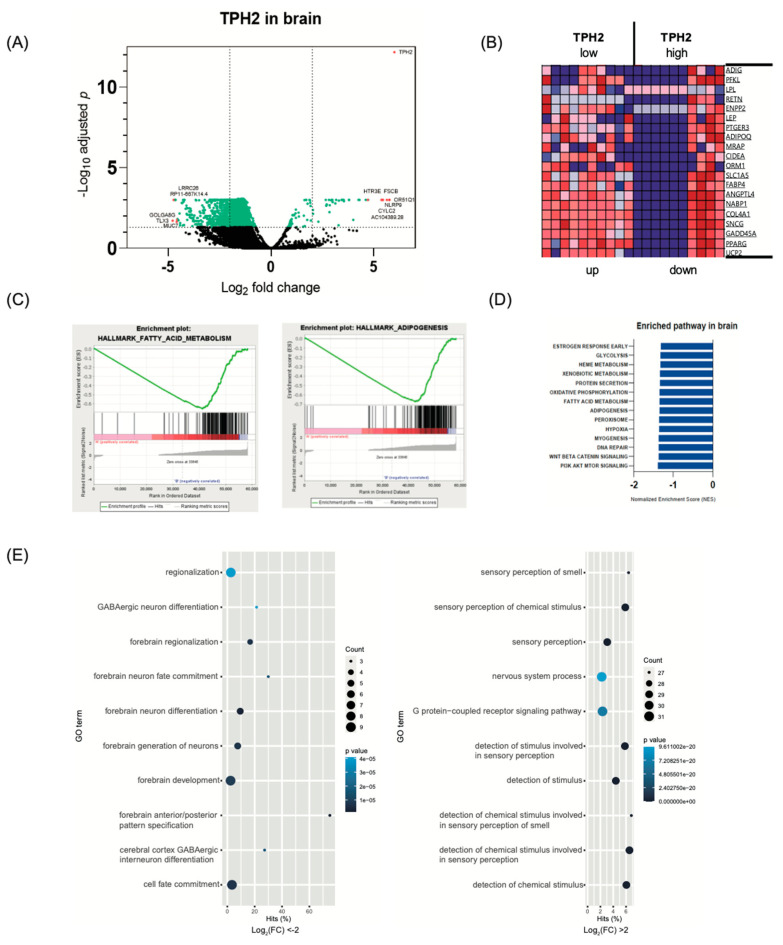
Gene expression changes in brain according to TPH2 expression (*n* = 10 per group). (**A**) Volcano plot of the significantly differentially expressed genes (DEGs). (**B**) Heatmap of DEGs. (**C**–**E**) DEGs were analyzed by Gene set enrichment analysis. (**C**) The enrichment plot for fatty acid metabolism and adipogenesis. (**D**) Bar plot depicting the normalized enrichment scores (NES). (**E**) Dot plot for enriched gene ontology pathways from GSEA results.

**Figure 3 ijms-22-02452-f003:**
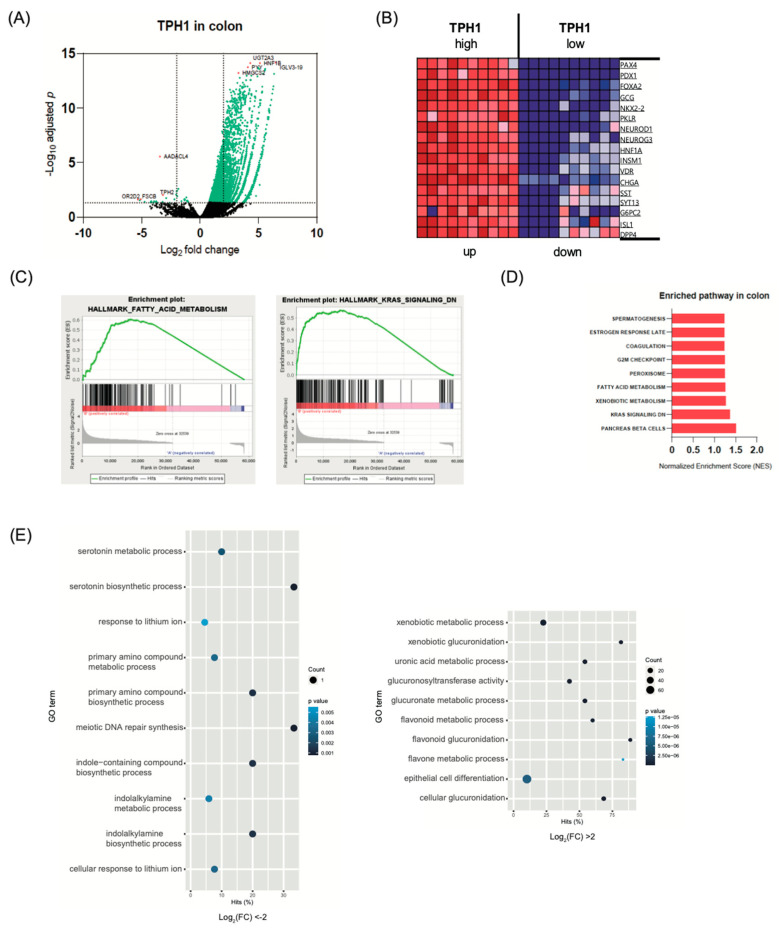
Gene expression changes in colon according to TPH1 expression (*n* = 10 per group). (**A**) Volcano plot of the significantly differentially expressed genes (DEGs). (**B**) Heatmap of DEGs. (**C**–**E**) DEGs were analyzed by Gene set enrichment analysis. (**C**) The enrichment plot for fatty acid metabolism and K-RAS signaling. (**D**) Bar plot depicting the normalized enrichment scores (NES). (**E**) Dot plot for enriched gene ontology pathways from GSEA results.

**Figure 4 ijms-22-02452-f004:**
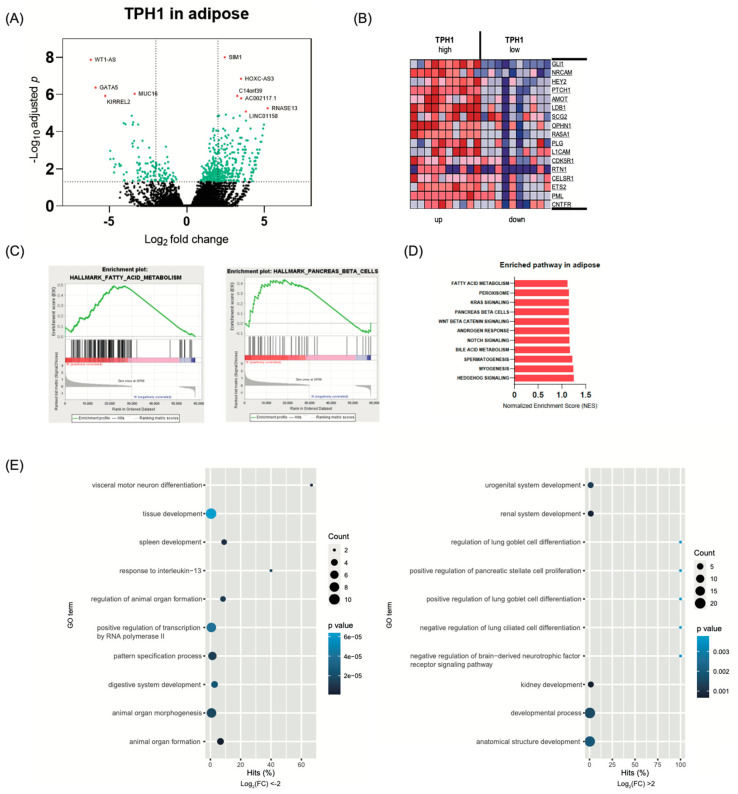
Gene expression changes in adipose tissue according to TPH1 expression (*n* = 10 per group). (**A**) Volcano plot of the significantly differentially expressed genes (DEGs). (**B**) Heatmap of DEGs. (**C**–**E**) DEGs were analyzed by gene set enrichment analysis. (**C**) The enrichment plot for fatty acid metabolism and pancreas beta cells. (**D**) Bar plot depicting the normalized enrichment scores (NES). (**E**) Dot plot for enriched gene ontology pathways from GSEA results.

**Figure 5 ijms-22-02452-f005:**
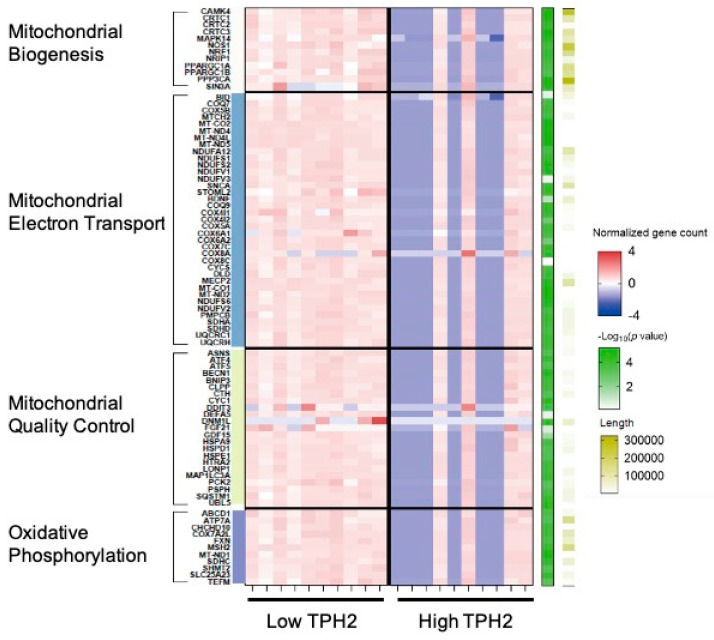
The heat map of DEGs related to mitochondrial biogenesis, energy metabolism, and mitochondrial quality control according to TPH2 expression in brain (*n* = 10 per group).

**Figure 6 ijms-22-02452-f006:**
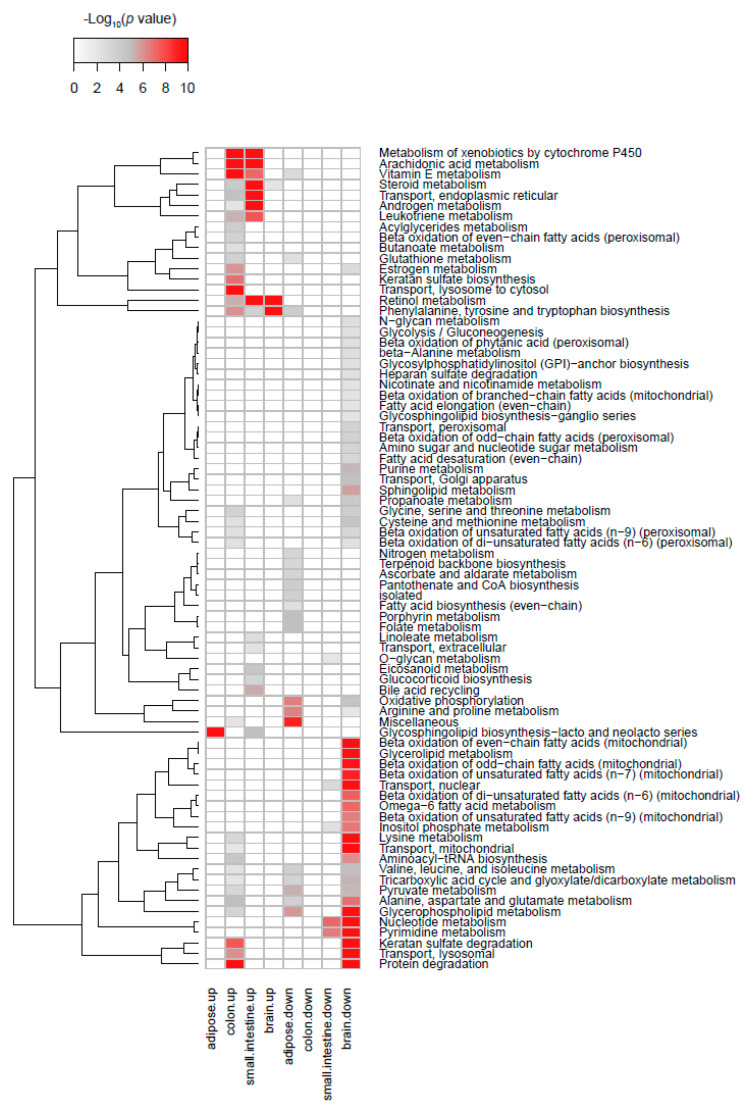
Predicted metabolic pathways according to TPH1 (adipose tissue, colon, and small intestine) and TPH2 (brain). Figure is the heatmap with the significant changes in metabolic pathway when the high TPH groups were compared with low TPH groups (*n* = 10 per each group). Up: upregulated pathways in high TPH tissue. Down: downregulated pathways in high TPH tissue.

## Data Availability

The data presented in this study are publicly available on the GTEx portal and HPA portal.

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
