# Peer review of "A Systems Biology Approach to Investigating the Interaction between Serotonin Synthesis by Tryptophan Hydroxylase and the Metabolic Homeostasis"

_ijms, 2021, doi:10.3390/ijms22052452_

Round 1

Reviewer 1 Report

This is an interesting study that attempts to elucidate the role of serotonin in metabolic homeostasis using a system biology approach with the GTEX, which is publicly available. The manuscript is reasonably well-written that clearly explained the rationale of the study conducted. However, there are several concerns:

  1. As the authors alluded to in the discussion, 5-HT levels in the tissue sites of interests were not available for analysis and as such, I feel the title of the manuscript, concerning serotonin, is somewhat misleading. The authors should be careful to not overstate the findings and the title should be changed to reflect the real subject of investigation ie how TPH1 and TPH2 mRNA levels are associated with the mRNA levels of genes implicated in various metabolic pathways. Whilst it is well-accepted that TPH is the rate-limiting enzyme of the serotonin biosynthesis process, are there any evidence showing that TPH overexpression results in high 5-HT levels? Wouldn't 5-HT levels also be dependent on substrate ie trytophan availability? This should be discussed. These are important issues to address as the finding of the current manuscript is largely based on the assumption that high TPH mRNA levels translate to high 5-HT levels, nowithstanding the fact that high mRNA levels may not even mean high TPH protein levels.
  2. What does the authors mean in this sentence in the introduction: "There is little evidence supporting the role of serotonin in human peripheral tissues."? What roles are the authors alluding to? If it is obesity and diabetes, associations between plasma serotonin levels, intestinal TPH1 levels and obesity in human have been reported by Young et al. in 2018 (https://doi.org/10.1038/s41366-018-0047-8) and differential glucose-responsiveness in 5-HT producing EC cells between lean and obese humans have been documented by the same group (https://doi.org/10.3390/nu11020234). Inclusion of the results from these studies will greatly strengthen the case for doing the current study.  
  3. How did the authors decide on 10 samples of the two ends of the TPH mRNA level spectrum? The authors should elaborate on what statistics was used to work out that a sample size of 10 in each high and low expression group has adequate power to detect the predefined effect size?
  4. A major concern for me is the validity of the study design. What is the range of TPH expression amongst each group eg. how high is the high TPH1 group and how low is the low TPH1 group? How much higher is the lowest in the high TPH1 group compared to the highest in the low TPH1 group? As these important details are not described, I do not even know if the TPH mRNA levels between the two groups are significantly different, which makes any downstream analyses and comparisons futile. 
  5. Figure 2-4 panels C-E: I am not a bioinformatics nor genetics expert and I really struggled to work out what are the authors trying to show in these figures. These images were of extremely low resolution and I cannot infer from these figures to verify what the authors are claiming ie "central serotonin has a negative association with fatty acid metabolism, adipogenesis and glycolysis." The authors should not assume that most readers are bioinformatics and genomics experts and as such, these graphs have to be much better annotated and explained in the results section.
  6. Figure 5 and corresponding results: The authors stated that brain TPH2 expression is associated with mitochondrial biogenesis etc. but did not elaborate on whether these are positive or negative associations. It will not be very informative to just know that there are associations without knowing the directions of the associations. 
  7. Discussion: as mentioned above, the authors should be careful not to overstate their findings. Nothing in the current study has anything to do with serotonin levels and such fact needs to be made clear in the discussion ie the phrase colonic TPH1 and adipocyte TPH1 expression should be used instead of gut-derived serotonin or adipocyte-derived serotonin. 
  8. Minor points:
    1. p2 of introduction: TPH2 is also expressed by pancreatic islets so the authors should rephrase the differences between TPH1 (peripheral) and TPH2 (central)
    2. Most of the body's circulating serotonin is stored in platelets, which is the gatekeeper of free circulating serotonin in plasma, the authors should at least mention this fact in the introduction and its potential implications.
    3. Results 2.2., second line "...brain tissues with highly expressed Tph1 mRNA..." should this have been TPH2? Also, by convention, TPH should be used instead of Tph when describing a human gene, please make sure this is consistent throughout the manuscript. 
    4. Discussion (p13): serotonin is not a protein. 

Author Response

This is an interesting study that attempts to elucidate the role of serotonin in metabolic homeostasis using a system biology approach with the GTEX, which is publicly available. The manuscript is reasonably well-written that clearly explained the rationale of the study conducted. However, there are several concerns:

  1. As the authors alluded to in the discussion, 5-HT levels in the tissue sites of interests were not available for analysis and as such, I feel the title of the manuscript, concerning serotonin, is somewhat misleading. The authors should be careful to not overstate the findings and the title should be changed to reflect the real subject of investigation ie how TPH1 and TPH2 mRNA levels are associated with the mRNA levels of genes implicated in various metabolic pathways. Whilst it is well-accepted that TPH is the rate-limiting enzyme of the serotonin biosynthesis process, are there any evidence showing that TPH overexpression results in high 5-HT levels? Wouldn't 5-HT levels also be dependent on substrate ie trytophan availability? This should be discussed. These are important issues to address as the finding of the current manuscript is largely based on the assumption that high TPH mRNA levels translate to high 5-HT levels, nowithstanding the fact that high mRNA levels may not even mean high TPH protein levels.

→ Thank you for your comments. As you pointed out, we don’t have 5-HT levels in the tissues and tryptophan availability is an important factor for the maintaining 5-HT levels in central and peripheral tissues. Thus, we changed our title as follows:

 “A systems biology approach to investigating the interaction between serotonin synthesis by tryptophan hydroxylase and metabolic homeostasis.”

→ Actually, several papers reported that tph mRNA levels have correlations with 5-HT levels (TPH overexpression results in high 5-HT levels)[1,2]. For example, beta cell specific Tph1 overexpressing transgenic rats shows serotonin increase in beta cell[1].

  1. What does the authors mean in this sentence in the introduction: "There is little evidence supporting the role of serotonin in human peripheral tissues."? What roles are the authors alluding to? If it is obesity and diabetes, associations between plasma serotonin levels, intestinal TPH1 levels and obesity in human have been reported by Young et al. in 2018 (https://doi.org/10.1038/s41366-018-0047-8) and differential glucose-responsiveness in 5-HT producing EC cells between lean and obese humans have been documented by the same group(https://doi.org/10.3390/nu11020234). Inclusion of the results from these studies will greatly strengthen the case for doing the current study.  

→ Thank you for your comments. We included these results in our manuscript as follows:

“Although some human studies reported that plasma 5-HT had associations with obesity [30, 31], more evidence is needed to support the role and potential mechanisms of 5-HT in human peripheral tissues.”

  1. How did the authors decide on 10 samples of the two ends of the TPH mRNA level spectrum? The authors should elaborate on what statistics was used to work out that a sample size of 10 in each high and low expression group has adequate power to detect the predefined effect size?

→ Thank you for your comments. Considering that GTEx dataset is from the normal population, it is generally believed that variations of gene expression are low. Therefore, it is important to stratify patients by remarkable gene expression changes, which led us to select a smaller number of samples (10 samples), but significant expression changes. Here we attached statistical significances of gene expression differences between high- and low-TPH groups using 10 samples and 50 samples, to speculate how number of samples affects the trends of gene expression changes (See table for rebuttal).

Figure X1. Log fold changes estimated from different number selected to define low and high TPH groups.

 Based on log-fold changes from different number of samples selected (i.e. 10 samples and 50 samples) (Figure X1 above), we saw significant correlated patterns of gene expression changes (Pearson’s correlation coefficients > 0.4; p-values < 10-15). Therefore, we concluded that our finding was not biased, but captured global trends from low- and high-TPH groups.

  1. A major concern for me is the validity of the study design. What is the range of TPH expression amongst each group eg. how high is the high TPH1 group and how low is the low TPH1 group? How much higher is the lowest in the high TPH1 group compared to the highest in the low TPH1 group? As these important details are not described, I do not even know if the TPH mRNA levels between the two groups are significantly different, which makes any downstream analyses and comparisons futile. 

→ Thank you for your nice comments. We added boxplots of TPH expressions of low- and high-TPH groups (Supplementary Figure 2) together statistical. We observed significant expression differences (t-test p-values < 0.01).

Supplementary Figure 3. Comparing Tph1/Tph2 expressions of tissues among low and high Tph groups

  1. Figure 2-4 panels C-E: I am not a bioinformatics nor genetics expert and I really struggled to work out what are the authors trying to show in these figures. These images were of extremely low resolution and I cannot infer from these figures to verify what the authors are claiming ie "central serotonin has a negative association with fatty acid metabolism, adipogenesis and glycolysis." The authors should not assume that most readers are bioinformatics and genomics experts and as such, these graphs have to be much better annotated and explained in the results section.

→ Thank you for your nice comments and really sorry for our images. As you pointed out, we improved our figure resolution.

  1. Figure 5 and corresponding results: The authors stated that brain TPH2 expression is associated with mitochondrial biogenesis etc. but did not elaborate on whether these are positive or negative associations. It will not be very informative to just know that there are associations without knowing the directions of the associations. 

→ Thank you for your comments. We used red and blue color to show relative expression of mitochondrial related genes in Figure 5 heatmap. Brain TPH2 groups shows negative associations with mitochondrial biogenesis. These results are about associations between TPH and mitochondrial genes. So TPH2 increase might be the protective response against to decreased mitochondrial biogenesis or increase 5-HT by TPH2 can decrease the mitochondrial biogenesis. Further studies are needed to elucidate causality in this association between TPH and mitochondrial. We changed our manuscript about Figure 5 as follows:

Result 2.5:

“In the brain, TPH2 expression showed negative associations with most genes related to mitochondrial biogenesis, ATP biosynthesis and mitochondrial quality control.”

→ And added more comments about this at discussion part as follows:

“Several studies have reported the protective role of 5-HT in mitochondrial dysfunction. 5-HT increased mitochondrial biogenesis in rodent cortical neurons [55] and mediates mitochondrial stress response in the neurons of C. elegans model [66]. Intriguingly, our data showed that high TPH2 group show lower expression of genes related to mitochondrial functions compared to low TPH2 group (Figure 5). This result implies TPH2 increase in brain could be the result of protective response against mitochondrial dysfunction. Further studies are needed to clarify this association between TPH2 and mitochondrial function.”

  1. Discussion: as mentioned above, the authors should be careful not to overstate their findings. Nothing in the current study has anything to do with serotonin levels and such fact needs to be made clear in the discussion ie the phrase colonic TPH1 and adipocyte TPH1 expression should be used instead of gut-derived serotonin or adipocyte-derived serotonin. 

→ Thank you for your comments. We changed some phrase as follows:

“TPH1 expressions in small intestine and colon have significant associations with hormone and nutrition regulation. This implies that GDS might be the major regulator of endocrine function and metabolic homeostasis in the gut. TPH1 expression in adipose tissue shows significant associations with mitochondrial FAO and development pathways of other tissues such as the lung and kidney.

  1. Minor points:
    1. p2 of introduction: TPH2 is also expressed by pancreatic islets so the authors should rephrase the differences between TPH1 (peripheral) and TPH2 (central)
    2. Most of the body's circulating serotonin is stored in platelets, which is the gatekeeper of free circulating serotonin in plasma, the authors should at least mention this fact in the introduction and its potential implications.
    3. Results 2.2., second line "...brain tissues with highly expressed Tph1 mRNA..." should this have been TPH2? Also, by convention, TPH should be used instead of Tph when describing a human gene, please make sure this is consistent throughout the manuscript. 
    4. Discussion (p13): serotonin is not a protein. 

→ Thank you for your comments.

8.1: As you pointed out, we rephrase the differences between TPH1 and TPH2 as follows:

“5-HT is synthesized from tryptophan catalyzed by the rate-limiting enzyme, tryptophan hydroxylase (TPH)[7]. TPH exists in two isoforms, TPH1 and TPH2. TPH1 is mainly expressed in peripheral tissues and pineal gland[8]. TPH2 is abundant in the serotonergic neurons in the brain and myenteric plexus[9].”

8.2: As you pointed out, we added some explanations about platelets as follows:

“More than 90% of peripheral serotonin is synthesized and secreted from enterochromaffin cells in the gut and stored in platelets[7]. Platelets uptake 5-HT from the plasma and release 5-HT in response to specific conditions such as tissue injury and acute inflammation [16,17]. Furthermore ~ ”  

8.3: We changed Tph1 -> TPH2. Thank you very much.

8.4: We removed “protein”. Thank you very much. 

Reviewer 2 Report

This is a very informative and thesis producing analysis, which can lead to future research opportunities. Studies indicate that the activation of serotonin signaling can have beneficial or harmful effects depending on the tissue where it may be activated. The vague term of “association” is used very often throughout the results.  Appropriate use of direction is advised (i.e. increase or decrease).  While the association analyses provide new insight into possible serotonin-mediated regulation of metabolism, there are certain concerns that should be addressed prior to publication:

Major:

  • The title is misleading as the study investigates the role of Tph1 and Tph2 expression and not the role of serotonin. If the authors suggest that the mRNA expression of Tph1 and Tph2 is always directly related to 5-HT accumulation, then they should at least reference the established evidence (i.e. increased Tph1 mRNA in human gut is associated with increased 5-HT in circulation). Otherwise, we cannot disregard that increased mRNA expression could be a response to reduced 5-HT.
  • Results 2.2: There seems to be a typo in the first sentence: “…..highly expressed Tph1 (Tph2?) mRNA and brain tissues with lowly expressed Tph2 mRNA.....”. If this weren’t a typo, please justify why you compared these tissues but not high vs low Tph2.
  • Results 2.2: It would have been very informative to also compare high vs low Tph1 in the select brain tissues as well as high Tph1 vs. high Tph2. If both enzymes serve the same purpose of replenishing 5-HT, they should follow a similar pattern.  This can also be applied to mitochondrial analysis in Figure 5.
  • Results 2.2: Please avoid using highly vague descriptions such as lipid metabolism and fatty acid metabolism. Please provide the exact pathway (fatty acid uptake, efflux, complex lipid synthesis, oxidation, lipolysis etc).
  • Results 2.2: authors associate central serotonin with reduced “lipid metabolism”, adipogenesis and glycolysis. Much attention is given to lipid metabolism whereas glucose is the primary source of energy for neurons.  These findings imply that high serotonin levels deplete neuronal energy by inhibiting glycolysis. This should be noted and compared to previous findings (i.e. 5HT2C).
  • Sentences that reference #34 are very obscure. How do lipids play a nutritional signal in the brain, which regulates systemic metabolism? Do the authors refer to leptin signaling and satiety? (there is also a grammar error: our results implies).
  • Results 2.3: very interesting and informative information. Well-done.
  • Results 2.4: The adipose tissue analyses should have been split into two: white adipose tissue (omental) vs. thermogenic adipose tissue (subcutaneous or subclavicular). If thermogenic adipose tissue cannot be analyzed, then the association study should be limited to omental adipose tissue only. Otherwise, the completely different metabolic states of these tissue will convolute the findings. Omental adipose tissue is not oxidative, thermogenic adipose tissue is.  Main purpose of omental adipose tissue is storage.  This is not the case for brown of beige adipocytes.
  • Sim1 expression is very specific to the paraventricular hypothalamus. It is not clear what an association within the adipose tissue really means.
  • Results 2.5: Background information leading to results indicates that serotonin increases mitochondrial function. However, the results indicate that high Tph2 group is associated with reduced mitochondrial function. The text uses the word “associated” again, but doesn’t explain the complete reversal of observations compared to previous studies. Could the increased Tph2 expression be redundant or in response to low serotonin to begin with?
  • Results 2.5: Pathway groups in Figure 5 seems to be put together by an algorithm. It is not clear why certain subunits of the electron transport chain and atp synthases are classified in different bmitochondrial pathways. Subunits of ATP synthesis, mitochondrial electron transport and oxidative phosphorylation all belong the complexes 1-5. For example, subunits of Complex I are listed under different mitochondrial pathways. This subgrouping seem arbitrary. 
  • Results 2.6: In Figure 6, the prediction table doesn’t show much of a difference between high and low adipose tissue groups while the differences are obvious in other tissues.
  • Results 2.6: In figure 6, FAO is assumed to be the only action against lipid overload in the text. Omental tissue is not oxidative and does not have loads of mitochondria. Brown and beige adipose tissues do where FAO is used for thermogenesis. Because the tissue analysis does not compare thermogenic adipose tissue alone, one cannot make this assumption over FAO. The major mode of lipid off-load in omental adipose tissue is lipolysis, not FAO.
  • Discussion refers to the “protein” level of serotonin.

Minor:

Inconsistencies in the flow of the article:

  • It is not clear whether HTR2A inhibition is good or bad:

Page 2 paragraph 2: GDS promotes fatty liver via HTR2A, and its inhibition prevents fatty liver.

Page 2 paragraph 3: HTR2A mutations are associated with obesity and metabolic syndrome.

  • Page 2 paragraph 4: The sentence “ There is little evidence to support the role of serotonin in human peripheral tissue” is too strong, especially when it comes following a list of background information on GDS, ADS and pancreatic serotonin. Perhaps the authors can take advantage of the inconsistency in point #1 above to warrant further research into tissue-specific role of serotonin.

Author Response

<REVIEWER 2>

This is a very informative and thesis producing analysis, which can lead to future research opportunities. Studies indicate that the activation of serotonin signaling can have beneficial or harmful effects depending on the tissue where it may be activated. The vague term of “association” is used very often throughout the results.  Appropriate use of direction is advised (i.e. increase or decrease).  While the association analyses provide new insight into possible serotonin-mediated regulation of metabolism, there are certain concerns that should be addressed prior to publication:

Major:

  • The title is misleading as the study investigates the role of Tph1 and Tph2 expression and not the role of serotonin. If the authors suggest that the mRNA expression of Tph1 and Tph2 is always directly related to 5-HT accumulation, then they should at least reference the established evidence (i.e. increased Tph1 mRNA in human gut is associated with increased 5-HT in circulation). Otherwise, we cannot disregard that increased mRNA expression could be a response to reduced 5-HT.

→ Thank you for your comments. As you pointed out, we don’t have 5-HT levels in the tissues central and peripheral tissues. Thus, we changed our title as follows:

 “A systems biology approach to investigating the interaction between serotonin synthesis by tryptophan hydroxylase and metabolic homeostasis.”

  • Results 2.2: There seems to be a typo in the first sentence: “…..highly expressed Tph1 (Tph2?) mRNA and brain tissues with lowly expressed Tph2 mRNA.....”. If this weren’t a typo, please justify why you compared these tissues but not high vs low Tph2.

→ Thank you for your comments. We changed Tph1 -> TPH2 in Results 2.2.

  • Results 2.2: It would have been very informative to also compare high vs low Tph1 in the select brain tissues as well as high Tph1 vs. high Tph2. If both enzymes serve the same purpose of replenishing 5-HT, they should follow a similar pattern.  This can also be applied to mitochondrial analysis in Figure 5.

→ It was our error (Tph1 in Result 2.2). We compared TPH2 in brain. Actually, we have same idea about your suggestion: comparing the role of TPH1 and TPH2. If we can dissect the role of TPH1 and TPH2 in brain, it will be the very informative work. This will be our next project. Thank you very much for your suggestion.

  • Results 2.2: Please avoid using highly vague descriptions such as lipid metabolism and fatty acid metabolism. Please provide the exact pathway (fatty acid uptake, efflux, complex lipid synthesis, oxidation, lipolysis etc).
  • Results 2.2: authors associate central serotonin with reduced “lipid metabolism”, adipogenesis and glycolysis. Much attention is given to lipid metabolism whereas glucose is the primary source of energy for neurons.  These findings imply that high serotonin levels deplete neuronal energy by inhibiting glycolysis. This should be noted and compared to previous findings (i.e. 5HT2C).

→ Thank you for your comments. As you pointed out, “lipid metabolism” is not “exact pathway”. So, we added more explanations in Result 2.2 as follows:

“For example, adipogenesis marker genes (FAB4 and ADIPOQ) and driver gene (PPARG) are decreased in high TPH2 group [40]. Fatty acid oxidation (FAO) related genes (CIDEA, UCP2, ANGPTL4) are also decreased in high TPH2 group [41].”

→ As you pointed out, this negative association may imply serotonin inhibit glycolysis. However, increased TPH2 expression also can be the response against reduced glycolysis. Association does not imply causation. Serotonin injection in rat increased anaerobic glycolysis in brain [3] and serotonin activates glycolysis in breast cancer cell lines[4]. Unfortunately, we cannot predict causality from DEG analysis or GSEA. This is the limitation of our analysis. So, we discussed about this in discussion part as follows:

“Further studies are needed to confirm this result by direct measurement of metabolites in human tissues”

  • Sentences that reference #34 are very obscure. How do lipids play a nutritional signal in the brain, which regulates systemic metabolism? Do the authors refer to leptin signaling and satiety? (there is also a grammar error: our results implies).

→ Thank you for your comments. In the brain, hypothalamus regulates nutritional status by sensing intracellular lipid levels. In the Referece#34, authors mentioned about the role of lipids as a nutritional signal in the brain as follows:

“AMP-activated protein kinase may also act as a cellular energy sensor within neurons to link neuronal lipid metabolism to systemic lipid metabolism and energy balance. AMPK is widely expressed in the ARC, PVN, and VMH of the hypothalamus and is able to sense intracellular energy status by the AMP/ATP ratio and the level of adipokines (e.g., leptin and ghrelin)”

“Importantly, the liver isoform of CPT-1, CPT-1a, is prevalent in the hypothalamus, and inhibition of hypothalamic CPT-1a causes an increase in intracellular LCFA-CoA, which triggers a satiation signal, leading to reduced systemic glucose production and food intake”

→ However, our sentence is obscure. So, we deleted this sentence about nutritional signal” and we changed “implies” -> “imply”. Thank you very much.

  • Results 2.3: very interesting and informative information. Well-done.

→ Thank you very much.

  • Results 2.4: The adipose tissue analyses should have been split into two: white adipose tissue (omental) vs. thermogenic adipose tissue (subcutaneous or subclavicular). If thermogenic adipose tissue cannot be analyzed, then the association study should be limited to omental adipose tissue only. Otherwise, the completely different metabolic states of these tissue will convolute the findings. Omental adipose tissue is not oxidative, thermogenic adipose tissue is.  Main purpose of omental adipose tissue is storage.  This is not the case for brown of beige adipocytes.

→ Thank you very much. As you pointed out, subcutaneous tissue has thermogenic tissue characteristics. But some papers reported that visceral adipose tissues also have thermogenic activity[5,6]. And subcutaneous tissue also acts as an energy storage organ like omental adipose tissue. Actually, subcutaneous tissue mainly acts an energy storage organ in general condition (under ambient temperature) and acts as a thermogenic organ in specific condition such as cold stimulation[7]. Thus, we consider omental and subcutaneous adipose tissue as “white adipose tissue”.

→ As you commented, we should not use “adipose tissue” in Result 2.4. Because we didn’t analyze brown adipose tissue. So, we changed the title of Result 2.4 and some words in discussion part and added some sentences about this problem in the discussion as follows:

2.4 Transcriptome analysis in white adipose tissue according to TPH1 expression”

TPH1 expression in white adipose tissue shows significant associations with mitochondrial FAO and development pathways of other tissues such as the lung and kidney. These results indicate that ADS is a critical factor for endocrine function in white adipose tissue.”

“Third, we used both omental adipose tissues and subcutaneous adipose tissues when we analyzed white adipose tissue transcriptome. This heterogeneity may act as a confounding factor[81].”

  • Sim1 expression is very specific to the paraventricular hypothalamus. It is not clear what an association within the adipose tissue really means.

→ Thank you very much for your comment. However, we found Sim1 expression in adipose tissue in human protein atlas database as follows(https://www.proteinatlas.org/ENSG00000112246-SIM1/tissue):

  • Results 2.5: Background information leading to results indicates that serotonin increases mitochondrial function. However, the results indicate that high Tph2 group is associated with reduced mitochondrial function. The text uses the word “associated” again, but doesn’t explain the complete reversal of observations compared to previous studies. Could the increased Tph2 expression be redundant or in response to low serotonin to begin with?

→ Thank you for your comments. As you commented, brain TPH2 groups shows negative associations with mitochondrial biogenesis. These results are about associations between TPH and mitochondrial genes. TPH2 increase might be the protective response against to decreased mitochondrial biogenesis or increase 5-HT by TPH2 can decrease the mitochondrial biogenesis. Further studies are needed to elucidate causality in this association between TPH and mitochondrial. We changed our manuscript about Figure 5 as follows:

Result 2.5:

“In the brain, TPH2 expression showed negative associations with most genes related to mitochondrial biogenesis, ATP biosynthesis and mitochondrial quality control.”

→ And added more comments about this at discussion part as follows:

“Several studies have reported the protective role of 5-HT in mitochondrial dysfunction. 5-HT increased mitochondrial biogenesis in rodent cortical neurons [55] and mediates mitochondrial stress response in the neurons of C. elegans model [66]. Intriguingly, our data showed that high TPH2 group show lower expression of genes related to mitochondrial functions compared to low TPH2 group (Figure 5). This result implies TPH2 increase in brain could be the result of protective response against mitochondrial dysfunction. Further studies are needed to clarify this association between TPH2 and mitochondrial function.”

  • Results 2.5: Pathway groups in Figure 5 seems to be put together by an algorithm. It is not clear why certain subunits of the electron transport chain and atp synthases are classified in different bmitochondrial pathways. Subunits of ATP synthesis, mitochondrial electron transport and oxidative phosphorylation all belong the complexes 1-5. For example, subunits of Complex I are listed under different mitochondrial pathways. This subgrouping seem arbitrary. 

→ Thank you for your comments. As you commented, we rearranged gene list and re-make new figure as follows:

  • Results 2.6: In Figure 6, the prediction table doesn’t show much of a difference between high and low adipose tissue groups while the differences are obvious in other tissues.

→ Thank you for your comments. This might be the result of heterogeneity of our white adipose tissue. To confirm this, we need to directly measure metabolite from adipose tissues. Actually, we made a plan to direct measure metabolite of adipose tissue from human and mice. And, as you recommended, we made a plan to analyze white, beige and brown adipose tissue by serotonin expression using GTEx and other public datasets. This is our next project. Thank you very much.

  • Results 2.6: In figure 6, FAO is assumed to be the only action against lipid overload in the text. Omental tissue is not oxidative and does not have loads of mitochondria. Brown and beige adipose tissues do where FAO is used for thermogenesis. Because the tissue analysis does not compare thermogenic adipose tissue alone, one cannot make this assumption over FAO. The major mode of lipid off-load in omental adipose tissue is lipolysis, not FAO.

→ Thank you for your comments. However, many papers suggest the human visceral adipose tissues have thermogenic activity. Following figure is the human visceral adipose tissue image of pheochromocytoma patient[8]. Mouse visceral fat also showed browning after cold stimulation[9]. Thus, we think we can discuss about FAO of omental tissue.

  • Discussion refers to the “protein” level of serotonin.

→ Thank you for your comments. We deleted ‘protein’ in this sentence.

Minor:

Inconsistencies in the flow of the article:

  • It is not clear whether HTR2A inhibition is good or bad:

Page 2 paragraph 2: GDS promotes fatty liver via HTR2A, and its inhibition prevents fatty liver.

Page 2 paragraph 3: HTR2A mutations are associated with obesity and metabolic syndrome.

→ Thank you for your comments. We don’t have information about the functional changes of HTR2A mutation in human. GWAS studies just give us association between SNP and human phenotypes. In mice study, Htr2A inhibition prevents fatty liver. We think mouse data about Htr2a provide us the meaning of HTR2A mutation in human. HTR2A inhibition is good for fatty liver disease. However, we don’t have clinical evidence about the role of HTR2A in human obesity.

  • Page 2 paragraph 4: The sentence “ There is little evidence to support the role of serotonin in human peripheral tissue” is too strong, especially when it comes following a list of background information on GDS, ADS and pancreatic serotonin. Perhaps the authors can take advantage of the inconsistency in point #1 above to warrant further research into tissue-specific role of serotonin.

→ Thank you for your comments. We want to emphasize the little evidence in “human” study. We agreed to your concern about “strong”. So, we changed our comment as follows:

“Although some human studies reported that plasma 5-HT had associations with obesity [30, 31], more evidence is needed to support the role and potential mechanisms of 5-HT in human peripheral tissues.” 
